# A Secure IoT-Based Authentication System in Cloud Computing Environment

**DOI:** 10.3390/s20195604

**Published:** 2020-09-30

**Authors:** Hsiao-Ling Wu, Chin-Chen Chang, Yao-Zhu Zheng, Long-Sheng Chen, Chih-Cheng Chen

**Affiliations:** 1Department of Information Engineering and Computer Science, Feng Chia University, Taichung 40724, Taiwan; 590590@gmail.com (H.-L.W.); alan3c@gmail.com (C.-C.C.); 2Department of Computer Science, National Tsing Hua University, Hsinchu 30013, Taiwan; s107062653@m107.nthu.edu.tw; 3Department of Information Management, Chaoyang University of Technology, Taichung 41349, Taiwan; 4Information and Engineering College, Jimei University, Fujian 361021, China; ccc@gm.cyut.edu.tw; 5Department of Industrial Engineering and Management, Chaoyang University of Technology, Taichung 413310, Taiwan

**Keywords:** Internet of things (IoT), lightweight authentication, user anonymity, cloud computing

## Abstract

The Internet of Things (IoT) is currently the most popular field in communication and information techniques. However, designing a secure and reliable authentication scheme for IoT-based architectures is still a challenge. In 2019, Zhou et al. showed that schemes pro-posed by Amin et al. and Maitra et al. are vulnerable to off-line guessing attacks, user tracking attacks, etc. On this basis, a lightweight authentication scheme based on IoT is proposed, and an authentication scheme based on IoT is proposed, which can resist various types of attacks and realize key security features such as user audit, mutual authentication, and session security. However, we found weaknesses in the scheme upon evaluation. Hence, we proposed an enhanced scheme based on their mechanism, thus achieving the security requirements and resisting well-known attacks.

## 1. Introduction

With the rapid development of computer science and network technology, the concept of the Internet of Things (IoT) has become a hot topic for research. A scientist named Ashton introduced this concept in 1991. In IoT, numerous sensors have the capability of collecting data and communicating with each other or providing data for human beings through the Internet.

Therefore, technology can be widely used in the smart power grid, smart home, and other fields. In a smart grid, sensors monitor electric energy consumption and time-of-use rates for power stations. Then, the stations can optimize power supply. In the intelligent transportation system, sensors monitor traffic to optimize navigation. In the smart home, users can control, monitor, and access items remotely. Though IoT is close to our lives, it suffers from security challenges due to the wireless nature of the communication channel [1].

In order to protect against those security challenges in IoT, authentication is indispensable. Authentication guarantees that the messages received by the receiver are from a legal message sender. It serves as the first line of defense against potential attackers. Authentication is considered the key requirement for IoT [2]. The cryptography in authentication falls into two broad categories: symmetric encryption and asymmetric encryption. Common asymmetric encryption includes elliptic-curve cryptography (ECC) and RSA encryption.

Asymmetric encryption uses pairs of keys, i.e., public key and private key. Although, asymmetric encryption is generally considered to have higher security, it requires a higher computational cost. On the other hand, common symmetric encryption, e.g., the advanced encryption standard (AES) and data encryption standard (DES), use a shared key between two or more parties. Symmetric encryption has the advantages of low computational cost and fast encryption speed. Some authentication schemes have been recently presented by using asymmetric encryptions [3,4,5,6,7,8,9,10]. However, traditional asymmetric encryptions do not suit IoT devices due to limited resources of most IoT devices, which gives rise to lightweight authentication schemes [11,12,13,14,15,16,17,18,19,20,21].

To solve security disadvantages, many lightweight authentication schemes have been proposed. In 1981, Lamport [22] first suggested lightweight authentication using a password. The scheme also uses hash chains to go through unsafe communication channel for remote user authentication. However, the scheme relies on a password table, which makes it very easy to steal personal data. After that, many user authentications with a password and key negotiation techniques have been put forward [23,24,25,26,27,28,29,30]. In 2007, Liao et al. [31] proposed an authentication scheme based on a hash function for a multi-server environment. Further, Hsiang et al. [32] pointed out that Liao et al.’s scheme [31] is subject to multiple security threats, e.g., insider attack, masquerade attack, and user/server forgery attacks. Hsiang et al. [32] then proposed a new authentication scheme and claimed their scheme has fewer computations and higher security. In 2011, Sood et al. [33] proposed an authentication scheme using a dynamic identity for multi-server circumstances and criticized Hsiang et al.’s scheme [32] for having a wrong password change phase and not resisting replay and impersonation attacks. In the same year, Lee et al. [34] assessed Sood et al.’s programme [33] and concluded that it was not safe. In 2014, Xue et al. [35] pointed out that Lee et al.’s scheme [34] failed under the circumstances of pseudonym attack and offline password guessing attack. Later, Amin et al. [36] criticized the scheme in [35], saying that it lacked identity hiding features and could not resist offline password guessing attack. Recently, some authentication schemes are also used in vehicular ad-hoc networks (VANETs) [37,38,39,40] or smart grid [41]. It shows the universality of authentication. In 2019, Zhou and other [42] proposed their scheme based on a hash function and exclusive or operation of the two-factor authentication scheme, claiming their authentication scheme has been proven safe and could resist various attacks.

We reviewed the scheme of Zhou et al. [42] and pointed out the weaknesses such as the inability of replay attacks to achieve user anonymity and provide mutual authentication. We proposed an improved scheme that has a better balance between efficiency and security. Therefore, the scheme is more suitable for IoT based environment. The contribution of this paper is to enhance the resistance to replay attack, thus improving user anonymity and providing mutual authentication based on Zhou et al.’s scheme [42].

The rest of this article is arranged as follows: Section 2 provides an overview of Zhou et al. ‘s scheme, focusing on its registration and certification phases. Then, the security analysis of the scheme proposed by Zhou et al. [42] was conducted. Section 3 introduces the scheme we proposed. Safety analysis and performance evaluation are described in Section 4 and Section 5. Section 6 gives the conclusion.

## 2. Related Works

In Section 2.1., we will introduce the authentication scheme proposed by Zhou et al. [42]. In addition, we will present the security issues of Zhou et al.’s scheme in Section 2.2.

### 2.1. Review of Zhou et al.’s Scheme

Zhou et al.’s scheme is divided into three stages: registration, authentication, and password modification. Here, we introduce the first two phases.

#### 2.1.1. Registration Phase

There are two parts in this phase: user registration and cloud server registration.

##### User Registration

First, user *U_i_* selects four values (i.e., identity *ID*_i_, pseudo-identity *PID_i_*, password *PW_i_*, and a random number *b_i_* to calculate *HP_i_* = *h*(*PW_i_ ||b_i_*). The *U_i_* then sends the *ID*_i_ and *PID_i_* to the control server *CS*. When *CS* receives (*ID*_i_, *PID_i_*), CS will check whether or not *ID*_i_ is in the database. If not, *CS* uses secret key x to calculate *C_1_^*^* = *h*(*PID_i_*||*IDcs*||*x*) and *C_2_^*^* = *h*(*ID*_i_||*x*); otherwise, *CS* will stop the authentication. *CS* stores *ID*_i_ in its database and sends (*C_1_^*^*, *C_2_^*^*, *IDcs*) to *U_i_*. When *U_i_* receives (*C_1_^*^*, *C_2_^*^*, *IDcs*), *U_i_* calculates three values, *C*_1_ = *C*_1_*^*^*⊕*HP_i_*, *C*_2_ = *C*_2_*^*^*⊕*h*(*ID_i_*||*HP_i_*), and *C*_3_ = *b_i_*⊕*h*(*ID_i_*||*PW_i_*), then stores (*C*_1_, *C*_2_, *C*_3_, *PID_i_*, *ID_cs_*) in a smart card.

##### Cloud Server Registration

Cloud server *S_j_* sends (*SID_j_*, *PSID_j_*) to *CS*, where *SID_j_* is the identity of *S_j_* and *PSID_j_* is the pseudo-identity of *S_j_*. When *CS* receives (*SID_j_*, *PSID_j_*), *CS* calculates *B*_1_ = *h*(*PSID_j_*||*ID_cs_*||*x*) and *B*_2_ = *h*(*SID_j_*||*x*). Finally, *CS* stores *SID_j_* in a database and sends (*B*_1_, *B*_2_, *ID_cs_*) to *S_j_*, and *S_j_* stores (*B*_1_, *B*_2_, *SID_j_*, *PSID_j_*, *ID_cs_*) in a memory.

#### 2.1.2. Authentication Phase

When user *U_i_* wants to connect with a cloud server, the user will perform the following five steps with the cloud server (*S_j_*) and the control server (*CS*).

Step 1: User inputs his *ID_i_* and *PW_i_*. A smart card will select a random number *r_u_* and new pseudo-identity *PID_i_^new^*; then, it calculates *b_i_* = *C*_3_⊕*h*(*ID_i_*||*PW_i_*), *HP_i_* = *h*(*PW_i_*||*b_i_*), *C*_1_*^*^* = *C*_1_⊕*HP_i_*, and *C*_2_*^*^* = *C*_2_⊕*h*(*ID_i_*||*HP_i_*). The smart card then calculates *D*_1_ = *C*_1_*^*^*⊕*r_u_*, *D*_2_ = *h*(*r_u_*||*PID_i_*||*ID_cs_*)⊕*ID_i_*, *D*_3_ = *C*_2_*^*^*⊕*h*(*ID_i_*||*HP_i_*)⊕ *PID_i_^new^*⊕*h*(*r_u_*||*ID_i_*), and *D*_4_ = *h*(*ID_i_*||*PID_i_*||*PID_i_^new^*||*r_u_*||*D*_3_). *U_i_* sends the message *M*_1_ = {*PID_i_*, *D*_1_, *D*_2_, *D*_3_, *D*_4_} to *S_j_*.

Step 2: When *S_j_* receives *M*_1_, *S_j_* selects a new pseudo-identity *PSID_j_^new^* and a random number *r_s_* to calculate *D*_5_ = *B*_1_⊕*r_s_*, *D*_6_ = *h*(*r_s_*||*PSID_j_*||*ID_cs_*)⊕*SID_j_*, *D*_7_ = *B*_2_⊕*PSID_j_^new^*⊕*h*(*r_s_*||*PSID_j_*), and *D*_8_ = *h*(*SID_j_*||*PSID_j_*||*PSID_j_^new^*||*r_s_*||*D*_7_). *S_j_* sends the message *M*_2_ = {*M*_1_, *PSID_j_*, *D*_5_, *D*_6_, *D*_7_, *D*_8_} to *CS*.

Step 3: When *CS* receives *M*_2_, *CS* calculates *r_u_* = *D*_1_⊕*h*(*PID_i_*||*ID_cs_*||*x*), *ID_i_* = *D*_2_⊕*h*(*r_u_*||*PID_i_*||*ID_cs_*), and *PIDi^new^* = *D*_3_⊕*h*(*ID_i_*||*x*)⊕*h*(*r_u_*||*ID_i_*). *CS* checks whether *ID_i_* in the database and *D*_4_? = *h*(*ID_i_*||*PID_i_*||*PIDi^new^*||*r_u_*||*D*_3_). If *ID_i_* is in the database and *D*_4_ = *h*(*ID_i_*||*PID_i_*||*PIDi^new^*||*r_u_*||*D*_3_), it means that *CS* confirms *U_i_* is a legal user. Otherwise, the authentication process will be terminated. Then, *CS* calculates *r_s_* = *D*_5_⊕*h*(*PSID_j_*||*ID_cs_*||*x*), *SID_j_* = *D*_6_⊕*h*(*r_s_*||*PSID_j_*||*ID_cs_*), and *PSID_j_* = *D*_7_⊕*h*(*SID_j_*||*x*)⊕*h*(*r_s_*||*SID_j_*). *CS* checks whether *SID_j_* is in database and *D*_8_ = *h*(*SID_j_*||*PSID_j_*||*PSID_j_^new^*||*r_s_*||*D*_7_). If *SID_j_* is in the database and *D*_8_ = *h*(*SID_j_*||*PSID_j_*||*PSID_j_^new^*||*r_s_*||*D*_7_), it means that *CS* confirms the *S_j_* is legal. Then, *CS* selects a random number *r_cs_* to calculate the session key *SK* = *h*(*r_u_*⊕*r_s_*⊕*r_cs_*), *D*_9_ = *h*(*PSID_j_^new^*||*ID_cs_*||*x*)⊕*h*(*r_s_*||*PSID_j_^new^*), *D*_10_ = *h*(*PSID_j_^new^*||*r_s_*||*PSID_j_*)⊕(*r_u_*⊕*r_cs_*), *D*_11_ = *h*(*SK_cs_*||*D*_9_||*D*_10_||*h*(*SID_j_*||*x*)), *D*_12_ = *h*(*PID_i_^new^*||*ID_cs_*||*x*)⊕*h*(*r_u_*||*PID_i_^new^*), *D*_13_ = *h*(*PID_i_^new^*||*r_u_*||*PID_i_*)⊕(*r_s_*⊕*r_cs_*), and *D*_14_ = *h*(*SK_cs_*||*D*_12_||*D*_13_||*h*(*ID_i_*||*x*)). *CS* sends the message *M*_3_ = {*D*_9_, *D*_10_, *D*_11_, *D*_12_, *D*_13_, *D*_14_} to *S_j_*.

Step 4: When *S_j_* receives *M*_3_, *S_j_* calculates (*r_u_*⊕*r_cs_* = *D*_10_⊕*h*(*PSID_j_^new^*||*r_s_*||*PSID_j_*). Hence, *S_j_* can compute *SK* = *h*(*r_u_*⊕*r_s_*⊕*r_cs_*). Then, *S_j_* checks *D*_11_? = *h*(*SK_s_*||*D*_9_||*D*_10_||*B*_2_) to confirm that *CS* is a legal control server or not. If *CS* is a legal control server, *S_j_* calculates *B*_1_*^new^* = *D*_9_⊕*h*(*r_s_*||*PSID_j_^new^*), updates *B*_1_ and *PSID_j_* as *B*_1_*^new^* and *PSID_j_^new^* in memory. *S_j_* sends message *M*_4_ = {*D*_12_, *D*_13_, *D*_14_} to *U_i_*.

When *U_i_* receives *M*_4_, *U_i_* calculates (*r_s_*⊕*r_cs_*) = *D*_13_⊕*h*(*PID_i_^new^*||*r_u_*||*PID_i_*) and *SK* = *h*(*r_u_**⊕**r_s_**⊕**r_cs_*). Then, *U_i_* checks *D*_14_? = *h*(*SK_u_*||*D*_12_||*D*_13_||*C*_2_*^*^*) to confirm that *CS* is a legal control server or not. *U_i_* calculates *C*_1_*^new^* = *D*_12_⊕*h*(*r_u_*||*PID_i_^new^*)⊕*HP_i_*, updates *C*_1_ and *PID_i_* in memory to *C_1_^new^* and *PID_i_^new^*.

### 2.2. Analysis of Zhou et al.’s Scheme

We found three weaknesses in Zhou et al.’s scheme at the certification stage. First, Zhou et al.’s scheme cannot achieve mutual authentication. Second, Zhou et al.’s scheme cannot work against a replay attack. Third, Zhou et al.’s scheme cannot guarantee anonymity in the authentication phase.

#### 2.2.1. Zhou et al.’s Scheme Cannot Achieve Mutual Authentication

Mutual authentication refers to the mutual verification between two entities. In Zhou et al.’s scheme, *CS* verifies *U_i_* by checking *D*_4_? = *h*(*ID_i_*||*PID_i_*||*PID_i_^new^*||*r_u_*||*D*_3_) in Step 3 of the authentication phase. We know *D*_3_ = *C*_2_*^*^*⊕*h*(*ID_i_*||*HP_i_*)⊕*PID_i_^new^*⊕*h*(*r_u_*||*ID_i_*) and *C*_2_*^*^* = *h*(*ID_i_*||*x*) from Step 1 of the authentication phase and the user registration. When *CS* computes *D*_3_⊕*h*(*ID_i_*||*x*)⊕*h*(*r_u_*||*ID_i_*), *CS* only can obtain *h*(*ID_i_*||*HP_i_*)⊕*PID_i_^new^*, where the parameter *HP_i_* is only known by *U_i_*. *CS* cannot successfully calculate *PID_i_^new^* from *D*_3_⊕*h*(*ID_i_*||*x*)⊕*h*(*r_u_*||*ID_i_*), even if the message *M*_1_ = {*PID_i_*, *D*_1_, *D*_2_, *D*_3_, *D*_4_} is sent from a legal user *U_i_*. Therefore, Zhou et al.’s scheme was unable to complete mutual authentication.

#### 2.2.2. Zhou et al.’s Scheme Cannot Guarantee Anonymity in Authentication Phase

A solution that provides anonymity must ensure that no one except the server knows the user’s personal information. We assume that the attacker *U_A_* is a legitimate user. Hence, *U_A_* will obtain ( $\bar{C_{1}^{*}}$ = *h*(*PID_A_*||*ID_cs_*||*x*), $\bar{C_{2}^{*}}$
*= h*(*ID_A_*||*x*), *ID_cs_*) from *CS* in the user registration phase. Once *U_A_* intercepts the message *M*_1_ = {*PID_i_*, *D*_1_, *D*_2_, *D*_3_, *D*_4_} from *U_i_* and uses *PID_i_* as new pseudo-identity to restart an authentication session, *U_A_* can obtain the *ID_i_* of the user *U_i_*. Details of the process are as follows.

Step 1: First, *U_A_* chooses a random number *r_A_* to calculate $\bar{D_{1}}$ = *C*_1_*^*^*⊕*r_A_*, $\bar{D_{2}}$ = *h*(*r_A_*||*PID_A_*||*ID_cs_*)⊕*ID_A_*, $\bar{D_{3}}$ = *C*_2_*^*^*⊕*h*(*ID_A_*||*HP_A_*)⊕*PID_i_*⊕*h*(*r_A_*||*ID_A_*), and $\bar{D_{4}}$ = *h*(*ID_A_*||*PID_A_*||*PID_i_*||*r_u_*|| $\bar{D_{3}}$). *U_A_* sends the message $\bar{M_{1}}$ = {*PID_A_*, $\bar{D_{1}}$, $\bar{D_{2}}$, $\bar{D_{3}}$, $\bar{D_{4}}$} to *S_j_*.

Step 2: When *U_A_* receives $\bar{M_{4}}$ = { $\bar{D_{12}}$, $\bar{D_{13}}$, $\bar{D_{14}}$}, *U_A_* can compute *ID_i_* = *D*_2_⊕*h*(*D*_1_⊕ $\bar{D_{12}}$⊕*h*(*r_A_*||*PID_i_*) ||*PID_i_*||*ID_cs_*), where *D*_1_ = *h*(*PID_i_*||*ID_cs_*||*x*)⊕*r_u_*, *D*_2_ = *h*(*r_u_*||*PID_i_*||*ID_cs_*)⊕*ID_i_*, and $\bar{D_{12}}$ = h(PID_i_||ID_cs_||x)⊕h(r_A_||PID_i_).

Therefore, Zhou et al.’s scheme cannot guarantee anonymity in the authentication phase.

## 3. Proposed Scheme

After we reviewed the shortcomings of Zhou et al.’s scheme, an improved scheme is put forward. The improvements include registration, authentication, and password modification.

### 3.1. Notations

The following is the introduction to the notations that will be used in our scheme.

*U_i_* is the *i*th user.

*ID_i_* is the *i*th user’s identity.

*PW_i_* is the *i*th user’s password.

*n_i_* is a random number.

*CS* is the control server.

*PID_i_* is the *i*th user’s pseudo-identity.

*ID_cs_* is the control server’s identity.

*SID_j_* is the *j*th server’s identity.

*PSID_j_* is the *j*th server’s pseudo-identity.

*x* is the secret key of *CS*.

*h* () is a one-way hash function.

*r_u_, r_s,_ r_cs_* are the random numbers selected by *U_i_, S_j_*, and *CS*.

*SK_u_, SK_s,_ SK_cs_* are the session keys for *U_i_, S_j_*, and *CS*.

*M*_1_, *M*_2_, *M*_3_, *M*_4_ are the messages in the authentication.

### 3.2. Registration Phase

This phase is divided into two parts: user registration and cloud server registration. When a user or a cloud server wants to join this system, he/she must run this phase first. After the user and the cloud server successfully finish this phase, they can connect with each other to start the authentication phase.

#### 3.2.1. User Registration

User *U_i_* selects their own id *ID_i_*, password *PW_i_*, random number *n_i_*. He/she sends *ID_i_* to *CS* by the secure channel. When *CS* receives *ID_i_*, *CS* checks it for its validity. If it is invalid, *CS* will stop this phase; otherwise, *CS* selects a pseudo-identity *PID_i_* for *U_i_* and uses the secret key *x* to compute *A_i_ = h(PID_i_*||*ID_cs_*||*x*) and *B_i_ = h*(*ID_i_*||*x*). *CS* stores *ID_i_* in its database and sends (*A_i_*, *B_i_*, *PID_i_*, *ID_cs_*) to *U_i_* by the secure channel. Once *U_i_* obtains these parameters, *U_i_* calculates *C*_1_ = *A_i_*⊕*h*(*ID_i_*||*n_i_*), *C*_2_ = *B_i_*⊕*h*(*PW_i_*||*n_i_*), *C*_3_ = *n_i_*⊕*h*(*ID_i_*||*PW_i_*), and *C*_4_
*= h*(*ID_i_*||*PW_i_*||*n_i_*) and then stores (*C*_1_, *C*_2_, *C*_3_, *C*_4_, *PID_i_*, *ID_cs_*) in a smart card. The flowchart for user registration is shown in Figure 1.

#### 3.2.2. Cloud Server Registration

A cloud server *S_j_* sends its identity *SID_j_* and a pseudo-identity *PSID_j_* to *CS* by a secure channel. Then, *CS* uses the secret key *x* to compute *A_j_* = *h*(*PSID_j_*||*ID_cs_*||*x*) and *B_j_* = *h*(*SID_j_*||*x*), stores *SID_j_* in its database, and sends (*A_j_*, *B_j_*, *ID_cs_*) to *S_j_* by a secure channel. When *S_j_* receives these parameters, *S_j_* stores (*A_j_*, *B_j_*, *SID_j_*, *SPID_j_*, *ID_cs_*) in its memory. The flowchart of the cloud server registration phase is shown in Figure 2.

### 3.3. Authentication Phase

When the user *U_i_* needs to retrieve services from the cloud server *S_j_*, this authentication must start to make sure of the legitimacy of both the user and the cloud server. After the authentication phase is completed, the user will negotiate a session key *SK*. By this session key, *U_i_* can connect with *S_j_* securely. The processes of the authentication phase are shown as follows and Figure 3.

Step 1: When user *U_i_* attempts to connect to cloud server *S_j_*, he/she inserts the smart card into a reader machine and keys in *ID_i_* and *PW_i_*. Then, the smart card selects a random number *r_u_* and calculates *n_i_* = *C*_3_⊕*h*(*ID_i_*||*PW_i_*). Then, the smart card checks *h*(*ID_i_*||*PW_i_*||*n_i_*)? = *C*_4_ to verify the identity and password. If the verification passed, the smart card will calculate *A_i_* = *C*_1_⊕*h*(*ID_i_*||*n_i_*), *B_i_* = *C*_2_⊕*h*(*PW_i_*||*n_i_*), *D*_1_ = *A_i_*⊕*r_u_*, *D*_2_ = *h*(*r_u_*||*PID_i_*||*ID_cs_*)⊕*ID_i_*, and *D*_3_ = *h*(*ID_i_*||*PID_i_*||*r_u_*). Finally, the smart card sends *M*_1_ = {*PID_i_*, *D*_1_, *D*_2_, *D*_3_} to *S_j_*.

Step 2: When *S_j_* receives *M*_1_, *S_j_* selects a new pseudo-identity $PSID_{j}^{\prime }$ and a random number *r_s_* to calculate *D*_4_ = *A_j_*⊕*r_s_*, *D*_5_ = *h*(*r_s_*||*PSID_j_*||*ID_cs_*)⊕*SID_j_*, *D*_6_ = *B_j_*⊕ $PSID_{j}^{\prime }\;$⊕*h*(*r_s_*||*PSID_j_*), and *D*_7_ = *h*(*SID_j_*||*PSID_j_*|| $PSID_{j}^{\prime }\;$||*r_s_*||*D*_6_). Then, *S_j_* sends message *M*_2_ = {*M*_1_, *PSID_j_*, *D*_4_, *D*_5_, *D*_6_, *D*_7_} to *CS*.

Step 3: Once *CS* receives *M*_2_, *CS* uses the secret key *x* to compute *r_u_* = *D*_1_⊕*h*(*PID_i_*||*ID_cs_*||*x*) and *ID_i_* = *D*_2_⊕*h*(*r_u_*||*PID_i_*||*ID_cs_*) and then checks whether *ID_i_* is valid and *D*_3_? = *h*(*ID_i_*||*PID_i_*||*r_u_*) or not. If the *ID_i_* is in its database and *D*_3_ = *h*(*ID_i_*||*PID_i_*||*r_u_*), it means that *U_i_* is legal. For the cloud server *S_j_*, *CS* uses the sccret key *x* to compute *r_s_* = *D*_4_⊕*h*(*PSID_j_*||*ID_cs_*||*x*), *SID_j_* = *D*_5_⊕*h*(*r_s_*||*PSID_j_*||*ID_cs_*), $PSID_{j}^{\prime }\;$=*D*_6_⊕*h*(*SID_j_*||*x*)⊕*h*(*r_s_*||*SID_j_*), and then checks whether *SID_j_* is in the database and *D*_7_ = *h*(*SID_j_*||*PSID_j_*|| $PSID_{j}^{\prime }\;$||*r_s_*||*D*_6_). If both conditions hold, it means that *S_j_* is legal. The processes of authentication phase will be stopped when any verification is wrong; otherwise, *CS* selects a random number *r_cs_* to compute the session key *SK_cs_* = *h*(*r_u_*⊕*r_s_*⊕*r_cs_*) for this round. Subsequently, for *S_j_*, *CS* computes *D*_8_ = *h*( $PSID_{j}^{\prime }\;$||*ID_cs_*||*x*)⊕*h*(*r_s_*||$PSID_{j}^{\prime }\;$), *D*_9_ = *h*( $PSID_{j}^{\prime }\;$||*r_s_*||*PSID_j_*)⊕(*r_u_*⊕*r_cs_*), and *D*_10_ = *h*(*SK_cs_*||*D*_8_||*D*_9_||*h*(*SID_j_*||*x*)). For *U_i_*, *CS* selects a new pseudo-identity $PID_{i}^{\prime }$ to compute *D*_11_= $PID_{i}^{\prime }\;$⊕*h*(*ID_i_*||*x*)⊕*h*(*r_u_*||*ID_i_*), *D*_12_ = *h*($PID_{i}^{\prime }\;$||*ID_cs_*||*x*)⊕*h*(*r_u_*||$PID_{i}^{\prime }\;$), *D*_13_ = *h*($PID_{i}^{\prime }\;$||*r_u_*||*PID_i_*)⊕(*r_s_*⊕*r_cs_*), and *D*_14_ = *h*(*SK_cs_*||*D*_12_||*D*_13_||*h*(*ID_i_*||*x*)). Finally, *CS* sends the message *M*_3_ = {*D*_8_, *D*_9_, *D*_10_, *D*_11_, *D*_12_, *D*_13_, *D*_14_} to *S_j_*.

Step 4: While *S_j_* receives *M*_3_, *S_j_* uses $PSID_{j}^{\prime }$ and *r_s_* to extract (*r_u_*⊕*r_cs_*) from *D*_9_, i.e., *r_u_*⊕*r_cs_* = *D*_9_⊕*h*( $PSID_{j}^{\prime }\;$||*r_s_*||*PSID_j_*). Then, *S_j_* checks *D*_10_? = *h*(*SK_s_*||*D*_8_||*D*_9_||*B_j_*), where *SK_s_ = h*(*r_u_*⊕*r_s_*⊕*r_cs_*). If this equation holds, it means that *CS* is legal; otherwise, this authentication process will be terminated. *S_j_* continues to calculate $A_{j}^{\prime }\;$ = *D*_8_⊕*h*(*r_s_*|| $PSID_{j}^{\prime }\;$) and updates *A_j_* and *PSID_j_* as $A_{j}^{\prime }$ and $PSID_{j}^{\prime }$ in the memory. At the end of this step, *S_j_* sends the message *M*_4_ = {*D*_11_, *D*_12_, *D*_13_, *D*_14_} to *U_i_*.

Step 4: Once the smart card receives *M*_4_, the smart card uses *B_i_*, *r_u_*, and *ID_i_* to extract $PID_{i}^{\prime }$ and (*r_s_*⊕*r_cs_*) from *D*_11_ and *D*_13_, respectively, i.e., $PID_{i}^{\prime }\;$ = *B_i_*⊕*D*_11_⊕*h*(*r_u_*||*ID_i_*) and (*r_s_*⊕*r_cs_*) = *D*_13_⊕*h*( $PID_{i}^{\prime }\;$||*r_u_*||*PID_i_*). The smart card will check whether or not *D*_14_? = *h*(*SK_u_*||*D*_12_||*D*_13_||*B_i_*), where *SK_u_ = h*(*r_u_*⊕*r_s_*⊕*r_cs_*). If this equation holds, it means that *CS* is legal; otherwise, this authentication process will be terminated. The smart card uses the new pseudo-identity $PID_{i}^{\prime }$ to calculate $C_{1}^{\prime }\;$ = *D*_12_⊕*h*(*r_u_*|| $PID_{i}^{\prime }\;$)⊕*h*(*ID_i_*||*n_i_*) and updates *C*_1_ and *PID_i_* as $C_{1}^{\prime }$ and $PID_{i}^{\prime }$. Finally, the smart card sends *h*(*SK_u_*) to *S_j_*.

Step 5: When *S_j_* receives *h*(*SK_u_*), *S_j_* will check *h*(*SK_u_*)? = *h*(*SK_s_*). If *h*(*SK_u_*) = *h*(*SK_s_*), this means that they already correctly negotiate the session key.

### 3.4. Password Change Phase

If the user *U_i_* needs to change the password, you may need to start the password change phase. First, we assume that the smart card of *U_i_* contains ( $C_{1}^{\prime }\;$, *C*_2_, *C*_3_, *C*_4_*,*
$PID_{i}^{\prime }\;$*, ID_cs_*). The *U_i_* inserts the smart card into the card reader for key verification in identity *ID_i_* and the original password *PW_i_.* The smart card will calculate *n_i_* = *C*_3_⊕*h*(*ID_i_*||*PW_i_*) and check *h*(*ID_i_*||*PW_i_*||*n_i_*)? = *C*_4_. If the equation holds, *U_i_* can input the new password $PW_{i}^{\prime }$. The smart card calculates $C_{2}^{\prime }\;$ = *C*_2_⊕*h*(*PW_i_*||*n_i_*)⊕*h*( $PW_{i}^{\prime }\;$||*n_i_*), $C_{3}^{\prime }\;$ = *C*_3_⊕*h*(*ID_i_*||*PW_i_*)⊕*h*(*ID_i_*|| $PW_{i}^{\prime }\;$), and $C_{4}^{\prime }\;$ = *C*_4_⊕*h*(*ID_i_*||*PW_i_*||*n_i_*)⊕*h*(*ID_i_*|| $PW_{i}^{\prime }\;$||*n_i_*) and replaces (*C*_2_, *C*_3_, *C*_4_) with ($C_{2}^{\prime }\;$, $C_{3}^{\prime }\;$, $C_{4}^{\prime }\;$). Finally, there are ($C_{1}^{\prime }\;$, $C_{2}^{\prime }\;$, $C_{3}^{\prime }\;$, $C_{4}^{\prime }\;$, $PID_{i}^{\prime }\;$, *ID_cs_*) in the smart card, and *U_i_* can use the new password $PW_{i}^{\prime }$ to perform the authentication phase in the next round. The flowchart of password modification phase is shown in Figure 4.

## 4. Security Analysis

In this section, we will analyze nine fundamental security requirements in which an authentication scheme should be achieved.

### 4.1. Mutual Authentication

As we discussed in Section 2.2.1., mutual authentication means that the identities of the two entities should be recognized before they connect. In our scheme, *CS* can be mutually authenticated with *U_i_* and *S_j_*, respectively.

#### 4.1.1. CS Verifies the Identity of Ui through Checking D3? = h(IDi‖PIDi‖ru)

In the user registration phase, *CS* computes *A_i_* = *h*(*PID_i_*||*ID_cs_*||*x*) and *B_i_ = h*(*ID_i_*||*x*) for *U_i_*, and two parameters are only known by *CS* and *U_i_*. When *U_i_* uses *A_i_* to hide the random number *r_u_* in the authentication phase, i.e., *D*_1_ = *A_i_*⊕*r_u_*, *CS* can use *h*(*PID_i_*||*ID_cs_*||*x*) to extract *r_u_*. Finally, *CS* can verify the identity of *U_i_* by equation *D*_3_ = *h*(*ID_i_‖PID_i_‖r_u_*).

#### 4.1.2. CS Verifies the Identity of Sj through Checking D7? = h(SIDj‖PSIDj‖PSIDj’‖rs‖D6)

In the cloud server registration phase, *CS* computes *A_j_* = *h*(*PSID_j_*||*ID_cs_*||*x*) and *B_j_* = *h*(*SID_j_*||*x*) for *S_j_*, and two parameters are only known by *CS* and *S_j_*. When *S_j_* uses *A_j_* to hide the random number *r_s_* in the authentication phase, i.e., *D*_4_ = *A_j_*⊕*r_s_*, *CS* can use *h*(*PSID_j_*||*ID_cs_*||*x*) to extract *r_s_*. Finally, *CS* can verify the identity of *S_j_* by equation *D*_7_ = *h*(*SID_j_*‖*PSID_j_*‖*PSID_j_^’^*‖*r_s_*‖*D*_6_).

#### 4.1.3. Sj Verifies the Identity of CS through Checking D10? = h(SKs‖D8‖D9‖Bj)

Because *B_j_* is only shared between *S_j_* and *CS*, they only have the capability of computing *h*(*SK_s_‖D*_8_*‖D*_9_*‖B_j_*). Therefore, *S_j_* can verify the identity of *CS* by equation *D*_10_ = *h*(*SK_s_‖D*_8_*‖D*_9_*‖B_j_*).

#### 4.1.4. Ui Verifies the Identity of CS through Checking D14? = h(SKu‖D12‖D13‖Bi)

Because *B_i_* only shares between *U_i_* and *CS*, they only have the capability of computing *h*(*SK_u_‖D*_12_*‖D*_13_*‖B_i_*). Therefore, *U_i_* can verify the identity of *CS* by equation *D*_14_ = *h*(*SK_u_‖D*_12_*‖D*_13_*‖B_i_*).

### 4.2. Session Key for All Entities

In the authentication phase, *U_i_*, *S_j_*, and *CS* generate *r_u_*, *r_s_*, and *r_cs_*, respectively. In addition, *U_i_*, *S_j_*, and *CS* obtain (*r_s_*⊕*r_cs_*), (*r_u_*⊕*r_cs_*), and (*r_u_*, *r_s_*) from *D*_13_, *D*_9_, and (*D*_1_, *D*_4_), respectively. Therefore, all entities can compute one same session key *SK* = *SK_cs_* = *SK_s_* = *SK_u_* = (*r_u_*⊕*r_s_*⊕*r_cs_*) in one session.

### 4.3. User Anonymity

The attacker’s use of user anonymity means that the user *U_i_* cannot be identified through the messages in the communication session [43]. In our authentication phase, *U_i_*’s identity *ID_i_* is protected by a hash function *D*_2_ = *h*(*r_u_*||*PID_i_*||*ID_cs_*)⊕*ID_i_*. Therefore, if an attacker wants to obtain *U_i_*’s identity, he/she must compute *h*(*r_u_*||*PID_i_*||*ID_cs_*). However, he/she cannot acquire the *r_u_* because he/she does not have the secret key *x* of *CS* to derive *r_u_* from *D*_1_ = *A_i_*⊕*r_u_*, where *A_i_* = *h*(*PSID_j_*||*ID_cs_*||*x*). Even if the attacker is a legal user, he/she still cannot obtain *h*(*r_u_*||*PID_i_*||*ID_cs_*) by adopting the strategy shown in Section 2.2.2. Therefore, the attacker cannot identify *U_i_*’s identity; furthermore, it shows that our proposed scheme has user anonymity.

### 4.4. Resistance to Off-Line Guessing Attack

Off-line guesswork attacks happen when an attacker obtains all the information stolen from the user, pass through insecure channels, and store in smart CARDS. The attacker can use the information held to guess the user’s identity and password.

We assume that an attacker gets (*C*_1_, *C*_2_, *C*_3_, *C*_4_, *PID_i_*, *ID_cs_*) that is stored in the user *U_i_*’s smart card and all messages (*M*_1_, *M*_2_, *M*_3_, *M*_4_) that pass by a nonsecure channel in the last session. Then, the attacker wants to guess a pair (*ID_i_, PW_i_*) from information. He/she can use the equation *D*_2_ = *h*(*r_u_*||*PID_i_*||*ID_cs_*)⊕*ID_i_* to confirm her/his guess *ID_i_*. According to the above hypothesis, the attacker has *PID_i_* and *D*_2_ from *M*_2_; *ID_cs_* is from the smart card. Therefore, he/she needs to get *r_u_*. Then, *r_u_* can be derived by rearranging *D*_1_ = *A_i_*⊕*r_u_* to *r_u_* = *A_i_*⊕*D*_1_. However, the attacker cannot compute *A_i_* = *h*(*PSID_j_*||*ID_cs_*||*x*) without the secret key *x* of *CS*. Therefore, he/she cannot successfully guess *ID_i_*. In addition, *PW_i_* only appears on *C*_2_ = *h*(*ID_i_*||*x*)⊕*h*(*PW_i_*||*n_i_*), *C*_3_ = *n_i_*⊕*h*(*ID_i_*||*PW_i_*), and *C*_4_
*= h*(*ID_i_*||*PW_i_*||*n_i_*). If the attacker wants to guess it, he/she needs to obtain *ID_i_*, *x* or *n_i_* first. However, the attacker cannot extract those values from intercepted messages. Therefore, he/she cannot successfully guess *PW_i_*. The results show that the scheme can resist offline guessing attack.

### 4.5. Resistance to Insider Attack

An insider attack means that an attacker is an inside member of the company of *CS*. He has the right to access the data stored in the *CS*’s database, e.g., the registered users’ identities and passwords. Then, he/she can use the information to simulate a legitimate user or cloud server. In our proposed scheme, only *ID_i_* and *SID_j_* are stored in *CS* for registration. There is no any other information for authentication stored in *CS*, i.e., *A_i_*, *B_i_*, *A_j_*, *B_j_*. Therefore, even if the inside attacker accesses the database of *CS*, he/she only can obtain the identity *ID_i_* of *U_i_* and *SID_j_* of *S_j_*; besides, the inside attacker still cannot impersonate the user *U_i_* or the cloud server *S_j_*. Thus, the scheme is able to resist internal attack.

### 4.6. Resistance to Stolen Smart Card Attack

Stolen card attack points to an attacker who steals the user’s smart card and extracts data stored in a smart card. Then, he/she uses these data to impersonate the user whose smart card was stolen. Here, we assume that an attacker already extracts the data (*C*_1_, *C*_2_, *C*_3_, *C*_4_*, PID_i_, ID_cs_*) from user *U_i_*’s smart card. In our proposed scheme, if the attacker wants to impersonate user *U_i_*, he/she needs to perform the authentication phase. According to the description of Step 1 in Section 3.2., the attacker needs to key in the correct *ID_i_* and *PW_i_* for checking the equation *h*(*ID_i_*||*PW_i_*||*n_i_*)? = *C*_4_. However, he/she does not have *ID_i_* and *PW_i_*. Therefore, when the attacker initiates an authentication run, he/she cannot pass the check *h*(*ID_i_*||*PW_i_*||*n_i_*)? = *C*_4_ in this step, then his/her authentication process will be terminated. The results show that the scheme can resist the attack of stolen smart cards.

### 4.7. Resistance to De-Synchronization Attack

An anti-synchronization attack means that an attacker interrupts and modifies the response message from the control server during the authentication phase, so that the authentication data between the client and the database of the control server are not synchronized [44]. Then, even if he/she is a legitimate user passing through the controlled server, all future authentication processes will fail.

In our proposed scheme, only users’ identities are stored in the control server’s database. In addition, those identities will not be changed in any phases, i.e., the authentication and password change phases. For the user, data changes occurred in the authentication stage and the last step of the password change phase. However, password change only needs to be involved on the user side; thus, the attacker cannot interfere. In the last step of the authentication phase, the data in the user’s smart card will be updated (*C*_1_, *PID_i_*) to ($C_{1}^{\prime }\;$, $PID_{i}^{\prime }\;$) when authentication processes are successfully finished. If the update was interrupted, the user can still use the old data (*C*_1_, *PID_i_*) to run a successful authentication process. It can be concluded that the scheme can resist synchronous attack.

### 4.8. Resistance to Forgery Attack

Counterfeit attack points to the attacker in the session is sent to the user, the cloud server and control server message, then the receiver will believe these messages are sent from a legal user, a cloud server, or the control server.

In our scenario, if an attacker wants to forge a user Ui, he/she would need to forge a message M1 to pass the equation *D3?* = *h*(*IDi‖PIDi‖ru*). However, the attacker cannot forge *D*_1_ = *A_i_*⊕*r_u_* because *A_i_* = *h*(*PID_i_*||*ID_cs_*||*x*) contains the secret key *x* of a control server. If the attacker wants to forge a cloud server, he/she needs to fabricate two messages, *M*_2_ and *M*_4_. To pass the equation *D*_7_? = *h*(*SID_j_**‖PSID_j_‖PSID_j_^’^‖r_s_‖D*_6_) and *D*_14_? = *h*(*SK_u_‖D*_12_*‖D*_13_*‖B_i_*); however, he/she cannot forge *D*_4_ = *A_j_*⊕*r_s_*, *D*_6_ = *B_j_*⊕ $PSID_{j}^{\prime }\;$⊕*h*(*r_s_*||*PSID_j_*) and *D*_14_ = *h*(*SK_cs_*||*D*_12_||*D*_13_||*h*(*ID_i_*||*x*)) because *A_j_* and *B_j_* both contain the secret key *x* of control server. If the attacker wants to forge the control server, he/she needs to make up a message *M*_3_ to pass the equation *D*_10_? = *h*(*SK_s_‖D*_8_*‖D*_9_*‖B_j_*). However, he/she cannot forge *D*_8_ = *h*( $PSID_{j}^{\prime }\;$||*ID_cs_*||*x*)⊕*h*(*r_s_*|| $PSID_{j}^{\prime }\;$) and *D*_10_ = *h*(*SK_cs_*||*D*_8_||*D*_9_||*h*(*SID_j_*||*x*)) because those messages contain the secret key *x* of the control server. As a result, we provide a solution to staying away from forgery attacks.

### 4.9. Resistance to User Tracking Attack

In terms of user tracking attacks, when an attacker eavesdrops on the delivered messages in different sessions, and then the attacker can confirm that two messages are from a fixed user according to a stable pseudo-identity being used. In our proposed scenario, the user Ui’s pseudo-identity would change in different sessions. Therefore, the attacker cannot ensure that any two messages are from the same user. The results show that the scheme can resist the user tracking attack.

## 5. Performance Evaluation

In this section, we will present the schemes of Maitra et al. [45], Amin et al. [36], Zhou et al. [42], and the performance evaluation of our schemes. Four authentication schemes only use a one-way hash operation, exclusive or operation, and concatenate operation. By comparing the execution time of an exclusive or operation to that of a one-way hash function or a symmetric algorithm, we ignored the execution time of an exclusive or operation., We chose SHA-2(256 bits) and AES as one-way hash functions and symmetric encryption/decryption algorithms, two of which are the most commonly used encryption methods in secure communications.

Table 1, Table 2 and Table 3 show a comparison of the security properties, computation cost, and communication cost among four respective authentication schemes. In Table 1, “O” means that the scheme can achieve a security requirement or resist the attack; “X” means that the scheme cannot achieve a security requirement or resist the attack. In Table 2, “*T_h_*” is one computation time of one-way hash function operation, and “*T_s_*” is one computation time of symmetric encryption/decryption. The “*T_h_*” and “*T_s_*” s’ values are 0.00517 ms and 0.02148 ms, respectively according to Zhou et al. [42].

Table 2 shows that our proposed scheme is in the middle regarding calculating costs. However, it is important to consider the trade-off between security and efficiency when we were designing a secure communication scheme. As can be seen from Table 1, the scheme proposed by us has better security than other schemes. We also assessed the communication costs of our scheme and other schemes, as shown in Table 3. The communication costs are the bits of parameters which passed during authentication. The Figure 5 shows the bar chart of the comparison of total calculation cost. Our scheme gets more cost than Zhou et al.’s [42] because we add an additional step at the last of the authentication phase to achieve mutual authentication. We only calculate the communication cost in the login and authentication phases due to the use of fewer number of times in the registration phase and password change phase. Therefore, in terms of security and efficiency, we can argue that our proposed scheme is more suitable for the Internet of Things environment than other related schemes.

Note that the outputs of the one-way hash function and the AES algorithm are 256 bits, and identities, pseudo-identities, and random numbers are 128 bits.

## 6. Conclusions

In this paper, we demonstrated that Zhou et al.’s scheme is not fully secure. Mutual authentication and anonymity cannot be guaranteed in the authentication phase. Then, we designed a new certification scheme to compensate for Zhou et al.’s scheme. The proposed scheme can resist common attacks and provide important features such as user anonymity and mutual authentication. We also added a new parameter in the first step of the authentication phase; moreover, it can detect whether or not the input identity and password are right at an early stage. Improved IoT-based authentication for cloud computing is also proposed, and the performance evaluation results show that the scheme has acceptable computation and good security. Therefore, we believe that this authentication scheme is applicable to real-world IoT devices.

In the future, we will investigate how to apply our IoT-based authentication mechanism in different computing environments, such as mobile environment and grid computing environment, etc. Furthermore, we are investigating how to make our system lightweight so that it can be widely used in the mobile computing world.

## Figures and Tables

**Figure 1 sensors-20-05604-f001:**
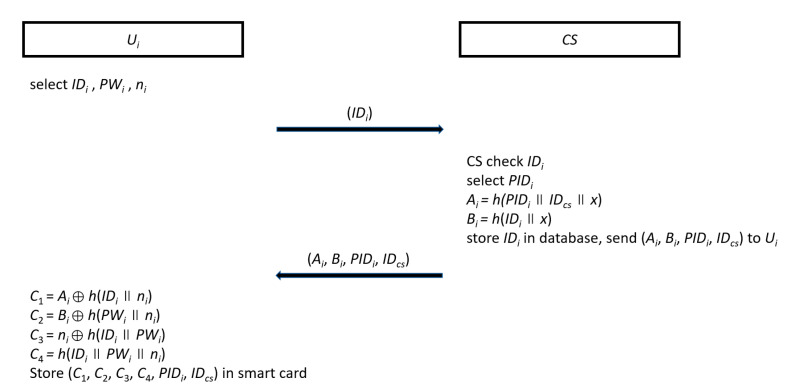
The flowchart of the user registration phase.

**Figure 2 sensors-20-05604-f002:**
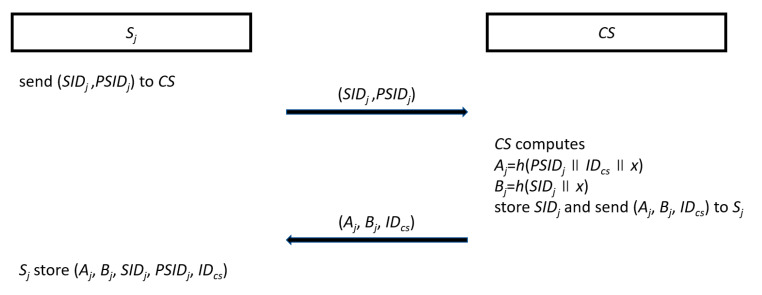
The flowchart of the cloud server registration phase.

**Figure 3 sensors-20-05604-f003:**
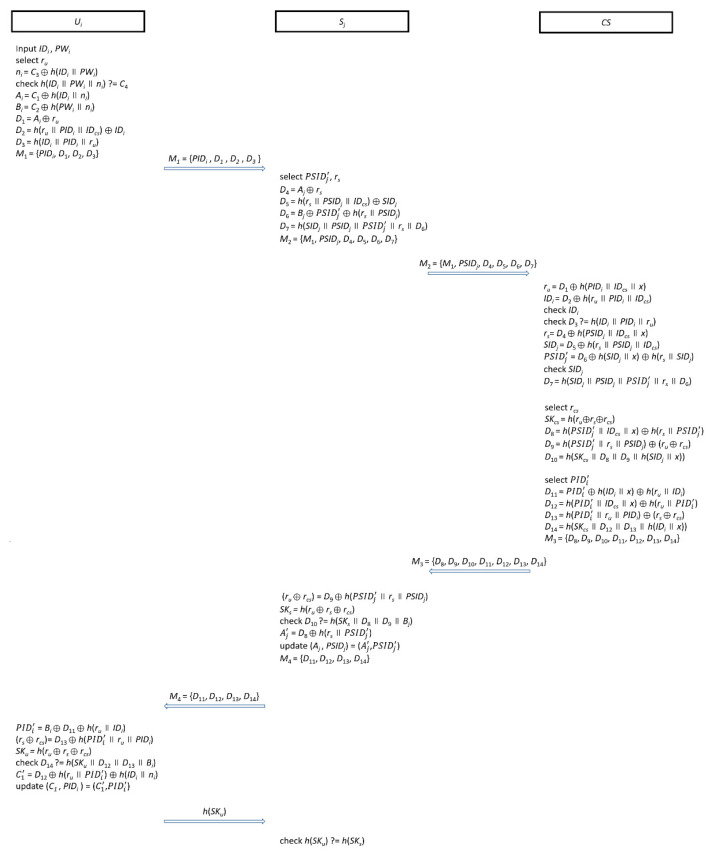
The processes of the authentication phase.

**Figure 4 sensors-20-05604-f004:**
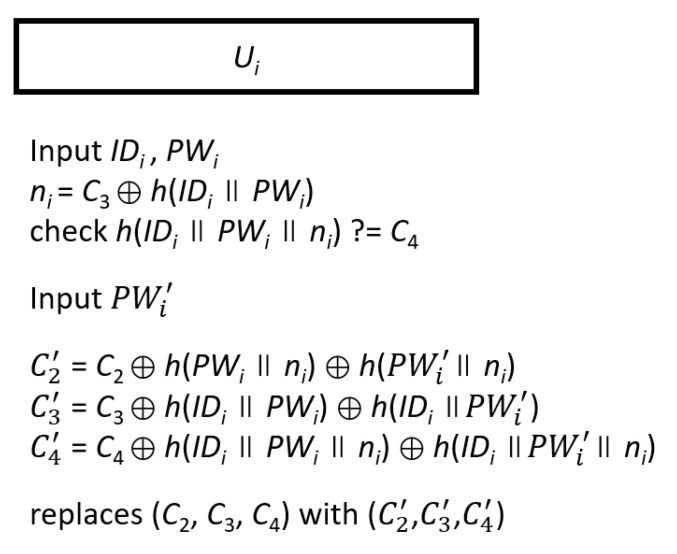
The flowchart of the password change phase.

**Figure 5 sensors-20-05604-f005:**
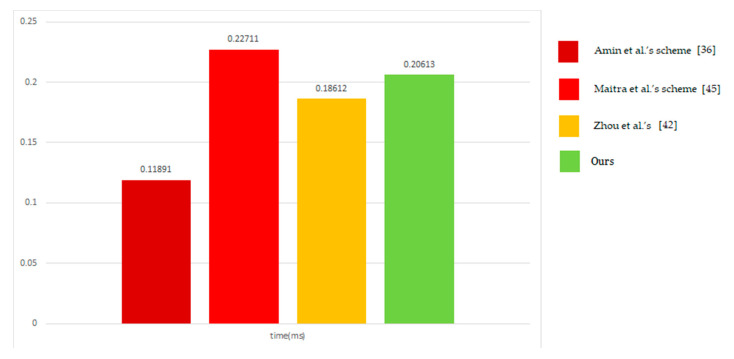
Comparison of total calculation cost (ms).

**Table 1 sensors-20-05604-t001:** Comparison of Security Properties among Four Authentication Schemes.

Property	R1	R2	R3	R4	R5	R6	R7	R8	R9
Amin et al.’s scheme [36]	O	O	O	X	O	O	O	O	X
Maitra et al.’s scheme [45]	O	X	O	X	O	O	O	O	X
Zhou et al.’s [42]	X	O	X	O	O	O	O	O	O
Ours	O	O	O	O	O	O	O	O	O

R1: Mutual authentication. R2: Session key for all entities. R3: User anonymity. R4: Resistance to off-line guessing attack. R5: Resistance to insider attack. R6: Resistance to stolen smart card attack. R7: Resistance to de-synchronization attack. R8: Resistance to forgery attack. R9: Resistance to user tracking attack.

**Table 2 sensors-20-05604-t002:** Calculation cost comparison of four certification schemes.

	Entities	Registration Phase	Login Phase	Authentication Phase	Password Change Phase	Total Operations of Login and Authentication
Amin et al.’s scheme [36]	U_i_	2 T_h_	6 T_h_	3 T_h_	7 T_h_	23 T_h_
	S_j_	0 T_h_	0 T_h_	4 T_h_	0 T_h_	
	CS	4 T_h_	0 T_h_	10 T_h_	0 T_h_	
Maitra et al.’s scheme [45]	U_i_	3 T_h_	6 T_h_	4 T_h_	9 T_h_	19 T_h_ + 6 T_s_
	S_j_	0 T_h_	0 T_h_ + 1 T_s_	4 T_h_ + 2 T_s_	0 T_h_	
	CS	3 T_h_ + 1 T_s_	0 T_h_	5 T_h_ + 3 T_s_	2 T_h_ + 2 T_s_	
Zhou et al.’s [42]	U_i_	3 T_h_	0 T_h_	10 T_h_	11 T_h_	36 T_h_
	S_j_	0 T_h_	0 T_h_	7 T_h_	0 T_h_	
	CS	4 T_h_	0 T_h_	19 T_h_	8 T_h_	
Ours	U_i_	4 T_h_	0 T_h_	12 T_h_	6 T_h_	39 T_h_
	S_j_	0 T_h_	0 T_h_	8 T_h_	0 T_h_	
	CS	4 T_h_	0 T_h_	19 T_h_	0 T_h_	

**Table 3 sensors-20-05604-t003:** Communication cost comparison of four authentication schemes.

Schemes	Communication Cost of L and A
Amin et al.’s scheme [36]	4736 bits
Maitra et al.’s scheme [45]	3072 bits
Zhou et al.’s [42]	5760 bits
Ours	6016 bits

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
