# Peer review of "A Secure IoT-Based Authentication System in Cloud Computing Environment"

_sensors, 2020, doi:10.3390/s20195604_

Round 1

Reviewer 1 Report

This paper has some issues:

  • Not innovative.
  • What sets are required for comparison. How can to ensure comparisons are fair and consistant.
  • More experiments and simulations are required befre comparison.
  • Too much emphasis on tehories. It lack the real orld solutions.
  • A professional English editing is required.

Author Response

Journal: Sensors

Manuscript ID: Sensos-920362

Type: Article   Number of Pages: 15

Title: A Secure IoT-based Authentication System in Cloud Computing Environment

Dear Editor,

Thank you very much for your letter and for the comments by the reviewers. These comments are very valuable and helpful for our paper.

We appreciate the careful, constructive, and generally favorable reviews given to our paper by the reviewers.

We believe we have adequately addressed all the excellent advices and questions raised by reviewers. Furthermore, we checked the manuscript and made sure the submitted manuscript is correct.

Please contact us if any further questions remain.

Sincerely yours,

Best regards.

Yours sincerely,

* Correspondence: Prof.  Chin-Chen Chang

Response to the comments of reviewers:

Reviewer 1:

Comments and Suggestions for Authors

Q 1 :This paper has some issues:Not innovative.What sets are required for comparison.

Ans: Thanks for the reviewer's suggestions.

The paper is about a lightweight authentication scheme based on IOT leveraging an authentication scheme. We proposed approach can resist various types of attacks and realise key security features such as user audit, mutual authentication and session security. This paper is technically sound and it represents a good and novel contribution to the knowledge body of this field.

In this paper, we demonstrated that Zhou et al. 's scheme is not fully secure. Mutual authentication and anonymity cannot be guaranteed in the authentication phase. Then, we designed a new certification scheme to compensate for Zhou et al’s scheme. The proposed scheme can resist common attacks and provide important features such as user anonymity and mutual authentication. We also added a new parameter in the first step of the authentication phase; moreover , it can detect whether or not  the input identity and password are right at an early stage. An improved IoT-based authentication for cloud computing is also proposed, and the performance evaluation results show that the scheme has acceptable computation and good security. Therefore, we believe that this authentication scheme is applicable to real-world IoT devices.

In the research gaps we encountered strike a balance between safety and efficiency. Security methods often require more calculations and communication. At present, we have adjusted to be more suitable for the IoT security communication protocol environment. The lightweight authentication scheme we also designed uses a small amount of calculation for higher security, which makes a more effective security protocol.

Q 2 :How can to ensure comparisons are fair and consistant.

Ans: Thanks for the reviewer's suggestions.

We have added a bar graph about the comparison of total calculation cost on line 382. It makes the evaluation more readable.

We also using better visual representation as Figure 5.Comparison of total calculation cost(ms) of the results can improve performance evaluation.

Figure 5. Comparison of total calculation cost(ms).

Q 3 :More experiments and simulations are required before comparison.

Ans: Thanks for the reviewer's suggestions.

We also have experiments and simulations as show Tables 1, 2, and 3 show a comparison of the security properties, computation cost, and communication cost among four respective authentication schemes. In Table 1, “O” means that the scheme can achieve a security requirement or resist the attack; “X” means that the scheme cannot achieve a security requirement or resist the attack. In Table 2, “Th” is one computation time of one-way hash function operation, and “Ts” is one computation time of symmetric encryption/decryption. The “Th” and “Ts” s’ values are 0.00517 ms and 0.02148 ms, respectively according to Zhou et al. [41].

Table 1. Comparison of Security Properties among Four Authentication Schemes.

property

R1

R2

R3

R4

R5

R6

R7

R8

R9

Amin et al.’s scheme [36]

O

O

O

X

O

O

O

O

X

Maitra et al.’s scheme [44]

O

X

O

X

O

O

O

O

X

Zhou et al.’s [41]

X

O

X

O

O

O

O

O

O

Ours

O

O

O

O

O

O

O

O

O

R1: Mutual authentication. R2: Session key for all entities. R3: User anonymity. R4: Resistance to off-line guessing attack. R5: Resistance to insider attack. R6: Resistance to stolen smart card attack. R7: Resistance to de-synchronization attack. R8: Resistance to forgery attack. R9: Resistance to user tracking attack.

Table 2 shows that our proposed scheme is in the middle  regarding calculating costs. However, it is important to consider the trade-off between security and efficiency when we were designing a secure communication scheme. As can be seen from Table 1, the scheme proposed by us has better security than other schemes. We also assessed the communication costs of our scheme and other schemes, as shown in Table 3. The communication costs are the bits of parameters which passed during authentication. Our scheme gets more cost than Zhou et al.’s [41] because we add an additional step at the last of the authentication phase to achieve mutual authentication. We only calculate the communication cost in the login and authentication phases due to the use of fewer number of times in the registration phase and password change phase. Therefore, in terms of security and efficiency, we can argue  that our proposed scheme is more suitable for the Internet of Things environment than other related schemes.

Table 2. Calculation cost comparison of four certification schemes.

Schemes

Entities

Registration phase

Login phase

Authentication phase

Password change phase

Total operations

of L and A

Amin et al.’s scheme [36]

Ui

2 Th

6 Th

3 Th

7 Th

23 Th

Sj

0 Th

0 Th

4 Th

0 Th

CS

4 Th

0 Th

10 Th

0 Th

Maitra et al.’s scheme [44]

Ui

3 Th

6 Th

4 Th

9 Th

19 Th +6 Ts

Sj

0 Th

0 Th +1 Ts

4 Th +2 Ts

0 Th

CS

3 Th +1 Ts

0 Th

5 Th +3 Ts

2 Th +2 Ts

Zhou et al.’s [41]

Ui

3 Th

0 Th

10 Th

11 Th

36 Th

Sj

0 Th

0 Th

7 Th

0 Th

CS

4 Th

0 Th

19 Th

8 Th

Ours

Ui

4 Th

0 Th

12 Th

6 Th

39 Th

Sj

0 Th

0 Th

8 Th

0 Th

CS

4 Th

0 Th

19 Th

0 Th

Table 3. Communication cost comparison of four authentication schemes.

Schemes

Communication cost of L and A

Amin et al.’s scheme [36]

4736 bits

Maitra et al.’s scheme [44]

3072 bits

Zhou et al.’s [41]

5760 bits

Ours

6016 bits

Note that the outputs of the one-way hash function and the AES algorithm are 256 bits, and identities, pseudo-identities, random numbers are 128 bits.

Q 4 :Too much emphasis on tehories. It lack the real orld solutions.

Ans: Thanks for the reviewer's suggestions.

After we reviewed the shortcomings of Zhou et al. 's scheme, an improved scheme is put forward. The improvements include registration, authentication, and password modification. In this paper, we demonstrated that Zhou et al. 's scheme is not fully secure. Mutual authentication and anonymity cannot be guaranteed in the authentication phase. Then, we designed a new certification scheme to compensate for Zhou et al’s scheme. The proposed scheme can resist common attacks and provide important features such as user anonymity and mutual authentication. We also added a new parameter in the first step of the authentication phase; moreover , it can detect whether or not  the input identity and password are right at an early stage. An improved IoT-based authentication for cloud computing is also proposed, and the performance evaluation results show that the scheme has acceptable computation and good security. Therefore, we believe that this authentication scheme is applicable to real-world IoT devices.

Q 5 :A professional English editing is required.

Ans: Thanks for pointing out the deficiency.

We invited a professional English revision agency to help us modify this article and revised part of the content is put in red and yellow.

Reviewer 2 Report

The paper is about a lightweight authentication scheme based on IOI leveraging an authentication scheme. The authors claim the proposed approach can resist various types of attacks and realise key security features such as user audit, mutual authentication and session security. The topic is on the current research hype about the integration of IoT and Cloud Computing environments. The paper is technically sound and it represents a good and novel contribution to the knowledge body of this field. Despite the fact the interest of the overall content, the paper suffers of organisation and presentation that should improved in order to mitigate the tough readability. The performance evaluation can be improved with a better visual representation of the results.

Author Response

Journal: Sensors

Manuscript ID: Sensos-920362

Type: Article   Number of Pages: 15

Title: A Secure IoT-based Authentication System in Cloud Computing Environment

Dear Editor,

Thank you very much for your letter and for the comments by the reviewers. These comments are very valuable and helpful for our paper.

We appreciate the careful, constructive, and generally favorable reviews given to our paper by the reviewers.

We believe we have adequately addressed all the excellent advices and questions raised by reviewers. Furthermore, we checked the manuscript and made sure the submitted manuscript is correct.

Please contact us if any further questions remain.

Sincerely yours,

Best regards.

Yours sincerely,

* Correspondence: Prof.  Chin-Chen Chang

Response to the comments of reviewers:

Reviewer 2:

Comments and Suggestions for Authors

Q 1 :

The paper is about a lightweight authentication scheme based on IOI leveraging an authentication scheme. The authors claim the proposed approach can resist various types of attacks and realise key security features such as user audit, mutual authentication and session security. The topic is on the current research hype about the integration of IoT and Cloud Computing environments. The paper is technically sound and it represents a good and novel contribution to the knowledge body of this field. Despite the fact the interest of the overall content, the paper suffers of organisation and presentation that should improved in order to mitigate the tough readability. The performance evaluation can be improved with a better visual representation of the results.

Ans: Thanks for the reviewer's suggestions. We have added a bar graph about the comparison of total calculation cost on line 382. It makes the evaluation more readable.

We also using better visual representation as Figure 5.Comparison of total calculation cost(ms) of the results can improve performance evaluation.

Figure 5. Comparison of total calculation cost(ms).

Reviewer 3 Report

In this paper, a lightweight authentication scheme based on IOI is proposed and improved. Thus, the security of the scheme is improved and the known attacks are resisted. However, there are still some problems to be improved.

  1. The manuscript has some grammatical or formatting errors. For example, on line 60, there is a problem with the format. And line 66, "change" should be changed to "Change". Please double check the manuscript carefully.
  2. The research question has not been clearly raised. Why propose an improved scheme to resist known attacks? Readers are not sure what specific problems the author wants to solve.
  3. These research gaps have not been well found or presented. How to solve these problems effectively?
  4. Section 2. The definition is not complete. Some parameters are not explained, for example, what is SID j? What is PSID j? Please check the other instructions carefully. And there is a problem with the format of the scheme.
  5. Section 3. It is suggested that the author divide Section 3.1 into section 3.1.1 and Section 3.1.
  6. Section 4. The font of Section 4.3 does not match the font format of other sections. Please check the manuscript carefully.
  7. Section 2. Some close related works, especially the new and important authentication schemes should be reviewed, such as “An efficient identity-based conditional privacy-preserving authentication scheme for vehicular ad hoc networks”.
  8. Section 5. According to the author's description, "O" and "X" in Table 1 indicate that the scheme can realize security requirements and resist attacks? The author does not explain the data in Table 3 very well. Why is the calculation cost of this scheme higher than other schemes? Can we improve the scheme and reduce the computational load?
  9. The title of the concluding section should be section 6. Please check the manuscript carefully. And the conclusion does not summarize the contribution of this paper.

Author Response

Journal: Sensors

Manuscript ID: Sensos-920362

Type: Article   Number of Pages: 15

Title: A Secure IoT-based Authentication System in Cloud Computing Environment

Dear Editor,

Thank you very much for your letter and for the comments by the reviewers. These comments are very valuable and helpful for our paper.

We appreciate the careful, constructive, and generally favorable reviews given to our paper by the reviewers.

We believe we have adequately addressed all the excellent advices and questions raised by reviewers. Furthermore, we checked the manuscript and made sure the submitted manuscript is correct.

Please contact us if any further questions remain.

Sincerely yours,

Best regards.

Yours sincerely,

* Correspondence: Prof.  Chin-Chen Chang

Response to the comments of reviewers:

Reviewer 3:

Comments and Suggestions for Authors

  3.In this paper, a lightweight authentication scheme based on IOI is proposed and improved. Thus, the security of the scheme is improved and the known attacks are resisted. However, there are still some problems to be improved.

Q 1 :

The manuscript has some grammatical or formatting errors. For example, on line 60, there is a problem with the format. And line 66, "change" should be changed to "Change". Please double check the manuscript carefully.

Ans: Thanks for the reviewer's suggestions. We have fixed this mistake, it is a mistake about the line up. It should be the“password change phase.”

In 2011, Sood et al. [33] proposed an authentication scheme using a dynamic identity for multi-server circumstances and criticized Hsiang et al.’s scheme [32] for having a wrong password change phase and not resisting replay and impersonation attacks. In the same year, Lee et al. [34] assessed Sood et al. 's programme [33] and concluded that it was not safe. In 2014, Xue et al. [35] pointed out that Lee et al. 's scheme [34] failed under the circumstances of pseudonym attack and offline password guessing attack.

Q 2 :

The research question has not been clearly raised. Why propose an improved scheme to resist known attacks? Readers are not sure what specific problems the author wants to solve.

Ans: Thanks for the reviewer's suggestions. We proposed an improved scheme that has a better balance between efficiency and security. Therefore, the scheme is more suitable for IoT based environment.

Q 3 :

These research gaps have not been well found or presented. How to solve these problems effectively?

Ans: Thanks for the reviewer's suggestions. The research gaps we encountered strike a balance between safety and efficiency. Security methods often require more calculations and communication. At present, we have adjusted to be more suitable for the IoT security communication protocol environment. The lightweight authentication scheme we designed uses a small amount of calculation for higher security, which makes a more effective security protocol.

Q 4 :

Section 2. The definition is not complete. Some parameters are not explained, for example, what is SID j? What is PSID j? Please check the other instructions carefully. And there is a problem with the format of the scheme.

Ans: Thanks for the reviewer's suggestions. We have added the definitions of SID and PSID in line101 on page 3.

Q 5 :

Section 3. It is suggested that the author divide Section 3.1 into section 3.1.1 and Section 3.1.

Ans: Thanks for the reviewer's suggestions. We have divided section 3.1 into section 3.1.1 and section 3.1.2.

3.1. Registration phase

This phase is divided into two parts: user registration and cloud server registration. When a user or a cloud server wants to join this system, he/she must run this phase first. After the user and the cloud server successfully finish this phase, they can connect with each other to start the authentication phase.

3.1.1. User registration

User Ui selects their own id IDi, password PWi, random number ni. He/she sends IDi to CS by the secure channel. When CS receives IDi, CS checks it for its validity. If it is invalid, CS will stop this phase; otherwise, CS selects a pseudo-identity PIDi for Ui and uses the secret key x to compute Ai=h(PIDi||IDcs||x) and Bi=h(IDi||x). CS stores IDi in its database and sends (Ai, Bi, PIDi, IDcs) to Ui by the secure channel. Once Ui obtains these parameters, Ui calculates C1= Aih(IDi||ni), C2=Bih(PWi||ni), C3=nih(IDi||PWi), and C4=h(IDi||PWi||ni) and then stores (C1, C2, C3, C4, PIDi, IDcs) in a smart card. The flowchart for user registration is shown in Figure 1.

Figure 1. The flowchart of the user registration phase.

3.1.2. Cloud server registration

A cloud server Sj sends its identity SIDj and a pseudo-identity PSIDj to CS by a secure channel. Then, CS uses the secret key x to compute Aj=h(PSIDj||IDcs||x) and Bj=h(SIDj||x), stores SIDj in its database, and sends (Aj, Bj, IDcs) to Sj by a secure channel. When Sj receives these parameters, Sj stores (Aj, Bj, SIDj, SPIDj, IDcs) in its memory. The flowchart of the cloud server registration phase is shown in Figure 2.

Figure 2. The flowchart of the cloud server registration phase.

Q 6 :

Section 4. The font of Section 4.3 does not match the font format of other sections. Please check the manuscript carefully.

Ans: Thanks for the reviewer's suggestions. We have fixed the font of section 4.3.

  1. Security Analysis

In this section, we will analyze  nine fundamental security requirements in which an authentication scheme should be achieved.

4.1. Mutual authentication

As we discussed in Section 2.2.1, mutual authentication means  that the identities of the two entities should be recognized before they connect. In our scheme, CS can be mutually authenticated with Ui and Sj, respectively.

.

.

4.3. User anonymity

The attacker's use of user anonymity means that the user Ui cannot be identified through the messages in the communication session [42]. In our authentication phase, Ui’s identity IDi is protected by a hash function D2= h(ru||PIDi||IDcs)⊕IDi. Therefore, if an attacker wants to obtain Ui’s identity, he/she must compute h(ru||PIDi||IDcs). However, he/she cannot  acquire  the ru because he/she does not have the secret key x of CS to derive ru from D1=Airu, where Ai=h(PSIDj||IDcs||x). Even if the attacker is a legal user, he/she still cannot obtain h(ru||PIDi||IDcs) by adopting the strategy shown in Subsection 2.2.2. Therefore, the attacker cannot identify Ui’s identity; furthermore, it shows that our proposed scheme has user anonymity.

Q 7 :

Section 2. Some close related works, especially the new and important authentication schemes should be reviewed, such as “An efficient identity-based conditional privacy-preserving authentication scheme for vehicular ad hoc networks”,” PA-CRT: Chinese Remainder Theorem Based Conditional Privacy-preserving Authentication Scheme in Vehicular Ad-hoc Networks”, “SPACF: A Secure Privacy-Preserving Authentication Scheme for VANET With Cuckoo Filter” and “EAAP: Efficient anonymous authentication with conditional privacy-preserving scheme for vehicular ad hoc networks”.

Ans: Thanks for the reviewer's suggestions. We have added these papers into the references.

  1. He, D.;Zeadally, S.; Xu, B.; Huang, X.; An Efficient Identity-Based Conditional Privacy-Preserving Authentication Scheme for Vehicular Ad Hoc Networks, IEEE Transactions on Information Forensics and Security. 2015, 10(12), 2681-2691, doi: 10.1109/TIFS.2015.2473820
  2. Zhang, J.;Cui, J.; Zhong, H.; Chen, Z.; Liu, L.; PA-CRT: Chinese Remainder Theorem Based Conditional Privacy-preserving Authentication Scheme in Vehicular Ad-hoc Networks, IEEE Transactions on Dependable and Secure Computing. 2019, Early Access ,1-1, doi: 10.1109/TDSC.2019.2904274
  3. Cui, J.;Zhang, J.; Zhong, H.; Xu, Y.; SPACF: A Secure Privacy-Preserving Authentication Scheme for VANET With Cuckoo Filter, IEEE Transactions on Vehicular Technology. 2017, 66(11), 10283-10295, doi: 10.1109/TVT.2017.2718101
  4. Azees, M.;Vijayakumar, P.; Deboarh, K.J.; EAAP: Efficient Anonymous Authentication With Conditional Privacy-Preserving Scheme for Vehicular Ad Hoc Networks, IEEE Transactions on Intelligent Transportation Systems. 2017, 18(9), 2467-2476, doi: 10.1109/TITS.2016.2634623

Q8 :

Section 5. According to the author's description, "O" and "X" in Table 1 indicate that the scheme can realize security requirements and resist attacks? The author does not explain the data in Table 3 very well. Why is the calculation cost of this scheme higher than other schemes? Can we improve the scheme and reduce the computational load?

The title of the concluding section should be section 6. Please check the manuscript carefully. And the conclusion does not summarize the contribution of this paper.

Ans: Thanks for the reviewer's suggestions.

The "X" means the scheme cannot realize security requirements and resist attacks. We have fixed this mistake.

In Table 1, “O” means that the scheme can achieve a security requirement or resist the attack; “X” means that the scheme cannot achieve a security requirement or resist the attack. In Table 2, “Th” is one computation time of one-way hash function operation, and “Ts” is one computation time of symmetric encryption/decryption. The “Th” and “Ts” s’ values are 0.00517 ms and 0.02148 ms, respectively according to Zhou et al. [41].

Table 1. Comparison of Security Properties among Four Authentication Schemes.

property

R1

R2

R3

R4

R5

R6

R7

R8

R9

Amin et al.’s scheme [36]

O

O

O

X

O

O

O

O

X

Maitra et al.’s scheme [44]

O

X

O

X

O

O

O

O

X

Zhou et al.’s [41]

X

O

X

O

O

O

O

O

O

Ours

O

O

O

O

O

O

O

O

O

The data in Table 3 are about communication cost. We added the explanation of the data and the reason of why our scheme has more cost.

 We also assessed the communication costs of our scheme and other schemes, as shown in Table 3. The communication costs are the bits of parameters which passed during authentication. Our scheme gets more cost than Zhou et al.’s [41] because we add an additional step at the last of the authentication phase to achieve mutual authentication. We only calculate the communication cost in the login and authentication phases due to the use of fewer number of times in the registration phase and password change phase. Therefore, in terms of security and efficiency, we can argue  that our proposed scheme is more suitable for the Internet of Things environment than other related schemes.

Table 3. Communication cost comparison of four authentication schemes.

Schemes

Communication cost of L and A

Amin et al.’s scheme [36]

4736 bits

Maitra et al.’s scheme [44]

3072 bits

Zhou et al.’s [41]

5760 bits

Ours

6016 bits

Note that the outputs of the one-way hash function and the AES algorithm are 256 bits, and identities, pseudo-identities, random numbers are 128 bits.

We have fixed the title of the concluding section and added the contribution of this paper has been added in the concluding section.

  1. Conclusions

In this paper, we demonstrated that Zhou et al. 's scheme is not fully secure. Mutual authentication and anonymity cannot be guaranteed in the authentication phase. Then, we designed a new certification scheme to compensate for Zhou et al’s scheme. The proposed scheme can resist common attacks and provide important features such as user anonymity and mutual authentication. We also added a new parameter in the first step of the authentication phase; moreover , it can detect whether or not  the input identity and password are right at an early stage. An improved IoT-based authentication for cloud computing is also proposed, and the performance evaluation results show that the scheme has acceptable computation and good security. Therefore, we believe that this authentication scheme is applicable to real-world IoT devices.

In the research gaps we encountered strike a balance between safety and efficiency. Security methods often require more calculations and communication. At present, we have adjusted to be more suitable for the IoT security communication protocol environment. The lightweight authentication scheme we also designed uses a small amount of calculation for higher security, which makes a more effective security protocol.

Reviewer 4 Report

In this paper, the authors pointed out security weakness of Zhou et al.’s authentication scheme for IoT and proposed an enhanced scheme that can achieve extra security. Through the performance evaluation, they showed that their scheme is superior to the other 3 schemes in terms of security. Having reviewed the paper, the following comments have been made:

1) In the evaluation, the computation cost is given based on th and ts units. The authors should consider giving some actual values for the units (e.g., like Zhou et al.’s paper), so that the results can be used in the selection of such schemes. Also, giving energy consumption due to the computation and communication overhead, if possible, would enhance the quality of the paper.

2) To prevent confusion, the author should consider the change of the term private key (x) since the term is used in asymmetric cryptographic systems.

3) Many editorial errors found. The paper should be rechecked thoroughly. For example, what is IOI in the abstract stands for? Some line breaks in the introduction section should be removed. In page 5, I think SPID in the body and Fig. 2 should be PSID. Check the format of subsection heads (e.g., subsect. 4.1 and 4.3).

Author Response

Journal: Sensors

Manuscript ID: Sensos-920362

Type: Article   Number of Pages: 15

Title: A Secure IoT-based Authentication System in Cloud Computing Environment

Dear Editor,

Thank you very much for your letter and for the comments by the reviewers. These comments are very valuable and helpful for our paper.

We appreciate the careful, constructive, and generally favorable reviews given to our paper by the reviewers.

We believe we have adequately addressed all the excellent advices and questions raised by reviewers. Furthermore, we checked the manuscript and made sure the submitted manuscript is correct.

Please contact us if any further questions remain.

Sincerely yours,

Best regards.

Yours sincerely,

* Correspondence: Prof.  Chin-Chen Chang

Response to the comments of reviewers:

Reviewer 4:

Comments and Suggestions for Authors

  1. In this paper, the authors pointed out security weakness of Zhou et al.’s authentication scheme for IoT and proposed an enhanced scheme that can achieve extra security. Through the performance evaluation, they showed that their scheme is superior to the other 3 schemes in terms of security. Having reviewed the paper, the following comments have been made:

Q1) In the evaluation, the computation cost is given based on th and ts units. The authors should consider giving some actual values for the units (e.g., like Zhou et al.’s paper), so that the results can be used in the selection of such schemes. Also, giving energy consumption due to the computation and communication overhead, if possible, would enhance the quality of the paper.

Ans: Thanks for the reviewer's suggestions. We have added the actual values of Th and Ts refer to Zhou et al.’s paper.

Tables 1, 2, and 3 show comparison of the security properties, computation cost, and communication cost among four authentication schemes, respectively. In Table 1, “O” means that the scheme can achieve a security requirement or resist the attack; “X” means that the scheme cannot achieve a security requirement or resist the attack. In Table 2, “Th” is one computation time of one-way hash function operation, and “Ts” is one computation time of symmetric encryption/decryption. The “Th” and “Ts” s’ values are 0.00517 ms and 0.02148 ms, respectively according to Zhou et al. [41].

Table 1. Comparison of Security Properties among Four Authentication Schemes.

property

R1

R2

R3

R4

R5

R6

R7

R8

R9

Amin et al.’s scheme [36]

O

O

O

X

O

O

O

O

X

Maitra et al.’s scheme [44]

O

X

O

X

O

O

O

O

X

Zhou et al.’s [41]

X

O

X

O

O

O

O

O

O

Ours

O

O

O

O

O

O

O

O

O

R1: Mutual authentication. R2: Session key for all entities. R3: User anonymity. R4: Resistance to off-line guessing attack. R5: Resistance to insider attack. R6: Resistance to stolen smart card attack. R7: Resistance to de-synchronization attack. R8: Resistance to forgery attack. R9: Resistance to user tracking attack.

Table 2 shows that our proposed scheme is in the middle in terms of calculating costs. However, it is important to consider the trade-off between security and efficiency when designing a secure communication scheme. As can be seen from Table 1, the scheme proposed by us has better security than other schemes. We also assess the communication costs of our scheme and other schemes, as shown in Table 3. The communication costs are the bits of parameters which passed during authentication. Our scheme gets more cost than Zhou et al.’s [41] because we add an additional step at the last of the authentication phase to achieve mutual authentication. We only calculate the communication cost in the login and authentication phases due to the use of fewer number of times in the registration phase and password change phase. Therefore, in terms of security and efficiency, we can say that our proposed scheme is more suitable for the Internet of Things environment than other related schemes.

Table 2. Calculation cost comparison of four certification schemes.

Schemes

Entities

Registration phase

Login phase

Authentication phase

Password change phase

Total operations

of L and A

Amin et al.’s scheme [36]

Ui

2 Th

6 Th

3 Th

7 Th

23 Th

Sj

0 Th

0 Th

4 Th

0 Th

CS

4 Th

0 Th

10 Th

0 Th

Maitra et al.’s scheme [44]

Ui

3 Th

6 Th

4 Th

9 Th

19 Th +6 Ts

Sj

0 Th

0 Th +1 Ts

4 Th +2 Ts

0 Th

CS

3 Th +1 Ts

0 Th

5 Th +3 Ts

2 Th +2 Ts

Zhou et al.’s [41]

Ui

3 Th

0 Th

10 Th

11 Th

36 Th

Sj

0 Th

0 Th

7 Th

0 Th

CS

4 Th

0 Th

19 Th

8 Th

Ours

Ui

4 Th

0 Th

12 Th

6 Th

39 Th

Sj

0 Th

0 Th

8 Th

0 Th

CS

4 Th

0 Th

19 Th

0 Th

Table 3. Communication cost comparison of four authentication schemes.

Schemes

Communication cost of L and A

Amin et al.’s scheme [36]

4736 bits

Maitra et al.’s scheme [44]

3072 bits

Zhou et al.’s [41]

5760 bits

Ours

6016 bits

Note that the outputs of the one-way hash function and the AES algorithm are 256 bits, and identities, pseudo-identities, random numbers are 128 bits.

Q2) To prevent confusion, the author should consider the change of the term private key (x) since the term is used in asymmetric cryptographic systems.

Ans: Thanks for the reviewer's suggestions. We have to change the term private key into a secret key.

Q3) Many editorial errors found. The paper should be rechecked thoroughly. For example, what is IOI in the abstract stands for? Some line breaks in the introduction section should be removed. In page 5, I think SPID in the body and Fig. 2 should be PSID. Check the format of subsection heads (e.g., subsect. 4.1 and 4.3).

Ans: Thanks for the reviewer's suggestions.

  1. We have fixed these mistakes. The “IOI” is actually “IoT”.

On this basis, a lightweight authentication scheme based on IoT is proposed, and an authentication scheme based on IoT is proposed, which can resist various types of attacks and realize key security features such as user audit, mutual authentication and session security.

2.There are no lines break in to introduction section now.

3.The SPID in Fig. 2 has been changed to PSID.

A cloud server Sj sends its identity SIDj and a pseudo-identity PSIDj to CS by a secure channel. Then, CS uses the secret key x to compute Aj=h(PSIDj||IDcs||x) and Bj=h(SIDj||x), stores SIDj in its database, and sends (Aj, Bj, IDcs) to Sj by a secure channel. When Sj receives these parameters, Sj stores (Aj, Bj, SIDj, PSIDj, IDcs) in its memory. The flowchart of the cloud server registration phase is shown in Figure 2.

4.The format mistake has been fixed.(e.g., subsect. 4.1 and 4.3).

4.1. Mutual authentication

As we discussed in Section 2.2.1, mutual authentication is that the identities of the two entities should be recognized before they connect. In our scheme, CS can be mutually authenticated with Ui and Sj, respectively.

4.3. User anonymity

The attacker's use of user anonymity means that the user Ui cannot be identified through the messages in the communication session [42]. In our authentication phase, Ui’s identity IDi is protected by a hash function D2= h(ru||PIDi||IDcs)⊕IDi. Therefore, if an attacker wants to obtain Ui’s identity, he/she must compute h(ru||PIDi||IDcs). However, he/she does not know the ru because he/she does not have the secret key x of CS to derive ru from D1=Airu, where Ai=h(PSIDj||IDcs||x). Even if the attacker is a legal user, he/she still cannot obtain h(ru||PIDi||IDcs) by adopting the strategy shown in Subsection 2.2.2. Therefore, the attacker cannot identify Ui’s identity; further, it shows our proposed scheme has user anonymity.

Round 2

Reviewer 1 Report

Authors should expand the content. References and comparisons can achieve this easier.

More literature and comparisons should be used. To help authors.

1) Advanced encryption for grid, IoT and big data can be used and compared.

A practical group blind signature scheme for privacy protection in smart grid. Journal of Parallel and Distributed Computing (2020)136, 29-39.

2) Ethical hacking and large scale tests should be undertaken t know vulnerabilities and improve.

3) Please proofread this paper, strengthen research contributions and highlight future direction more.

Author Response

We believe we have adequately addressed all the excellent advices and questions raised by reviewers. Furthermore, we checked the manuscript and made sure the submitted manuscript is correct.

Please contact us if any further questions remain.

Sincerely yours,

Best regards.

Yours sincerely,

* Correspondence: Prof.  Chin-Chen Chang

Response to the comments of reviewers:

Reviewer 1:

Comments and Suggestions for Authors

Authors should expand the content. References and comparisons can achieve this easier.

More literature and comparisons should be used. To help authors.

Q 1) Advanced encryption for grid, IoT and big data can be used and compared.

Ans: Thanks for your valuable suggestion. We have added the indicated literature in our introduction part and references. Since the IoT-based authentication system proposed in our manuscript is suitable for the cloud computing environment, our next step is to investigate our approach for different computing environments, such as grid computing. Then we can compare the performance of ours with some existing methods in the same environment.

At line 71 and 72, reference [41].

Recently, some authentication schemes are also used in Vehicular Ad-hoc Networks(VANETs) [37] [38] [39] [40] or smart grid [41].  It shows the universality of authentication. In 2019, Zhou and other [42]  proposed their scheme_based on a hash function and exclusive or operation of the two-factor authentication scheme,claiming their authentication scheme was proved to be safe and could resist various attacks.

  1. Conclusions

In the future, we will investigate how to apply our IoT-based authentication mechanism in different computing environments, such as mobile environment and grid computing environment, etc. Furthermore, we are investigating how to make our system lightweight so that it can be widely used in the mobile computing world.

References

  1. Kong, W.;Shen, J.; Vijayakumar, P.; Cho, Y.; Chang, V.; A practical group blind signature scheme for privacy protection in smart grid, Journal of Parallel and Distributed Computing. 2020, 136, 29-39. doi: 10.1016/j.jpdc.2019.09.016

Q 2) Ethical hacking and large scale tests should be undertaken know vulnerabilities and improve.

Ans: Thanks for your good suggestion. After we checked authentication schemes proposed previously, they didn’t test their methods by large scale tests, but just analyzed the fundamental security requirements. We have already shown that our method can successfully achieve nine fundamental security requirements in Section 4.

Our paper is about a lightweight authentication scheme based on IOT leveraging an authentication scheme. We proposed an approach that can resist various types of attacks and realize key security features such as user audit, mutual authentication, and session security. This paper is technically sound and it represents a good and novel contribution to the knowledge body of this field. In this paper, we demonstrated that Zhou et al. 's scheme is not fully secure. Mutual authentication and anonymity cannot be guaranteed in the authentication phase. Then, we designed a new certification scheme to compensate for Zhou et al’s scheme. The proposed scheme can resist common attacks and provide important features such as user anonymity and mutual authentication. We also added a new parameter in the first step of the authentication phase; moreover, it can detect whether or not the input identity and password are right at an early stage. Improved IoT-based authentication for cloud computing is also proposed, and the performance evaluation results show that the scheme has acceptable computation and good security. Therefore, we believe that this authentication scheme is applicable to real-world IoT devices. We designed 5 attack methods (R4- R9)in line 301 – 371 on pages 10-11. We also design a lightweight authentication scheme based on IOT leveraging an authentication scheme, which is not designed for commercial purpose. But we also design the  performance evaluation on section 5 and comparison table in Table 1.

Table 1. Comparison of Security Properties among Four Authentication Schemes.

property

R1

R2

R3

R4

R5

R6

R7

R8

R9

Amin et al.’s scheme [36]

O

O

O

X

O

O

O

O

X

Maitra et al.’s scheme [45]

O

X

O

X

O

O

O

O

X

Zhou et al.’s

X

O

X

O

O

O

O

O

O

Ours

O

O

O

O

O

O

O

O

O

R1: Mutual authentication. R2: Session key for all entities. R3: User anonymity.

R4: Resistance to off-line guessing attack. R5: Resistance to insider attack.

R6: Resistance to stolen smart card attack. R7: Resistance to de-synchronization attack. R8: Resistance to forgery attack. R9: Resistance to user tracking attack.

Q 3) Please proofread this paper, strengthen research contributions, and highlight future direction more.

 Ans: Thanks for pointing out the deficiency. We have already proofread our manuscript. Some merits of this manuscript are stated in conclusion and the future research directions.There are still some challenges in the way of IoT-based authentication for cloud computing. In the future, we can get a more efficient and feasible solution in the cloud computing environment. The revised part is marked in blue at line 421 – 423.

Reviewer 3 Report

The authors have improved the quality of the paper according to the reviewers' comments.

Author Response

期刊:傳感器

出版物ID:Sensos - 920362

類型:文章頁數:15   

標題:雲計算環境中基於物聯網的安全認證系統

親愛的編輯,

非常感謝您的來信和審稿人的評論。這些評論對我們的論文非常有價值且有幫助。

我們感謝審稿人對我們的論文進行了認真,建設性和總體上有利的審閱。

我們相信我們已經充分解決了審稿人提出的所有出色建議和問題。此外,我們檢查了稿件,並確保提交的稿件正確。

如果還有其他問題,請與我們聯繫。

您忠誠的,

最好的祝福。

此致,

*通訊:張進晨教授

對審稿人評論的回應:

評論者3:

給作者的評論和建議

 問:需要專業的英語編輯。

答:感謝您指出不足之處。

我們邀請了專業的英語修訂機構來幫助我們修改本文,並重新修訂後的部分內容用紅色和黃色表示。
